# Consumer Behaviour and Food Waste: Understanding and Mitigating Waste with a Technology Probe

**DOI:** 10.3390/foods11142048

**Published:** 2022-07-11

**Authors:** Eliot Jones-Garcia, Serafim Bakalis, Martin Flintham

**Affiliations:** 1Horizon Centre for Doctoral Training, School of Computer Science, University of Nottingham, Jubilee Campus, Wollaton Road, Nottingham NG8 1BB, UK; eliot.jones@nottingham.ac.uk; 2Future Food Beacon, University of Nottingham, Sutton Bonington Campus, Nottingham LE12 5RD, UK; 3Mixed Reality Lab, School of Computer Science, University of Nottingham, Jubilee Campus, Nottingham NG8 1BB, UK; martin.flintham@nottingham.ac.uk; 4Department of Food Science, University of Copenhagen, DK-1958 Copenhagen, Denmark

**Keywords:** smart bin, human computer interaction, internet of things, social practice

## Abstract

Globally, nearly one third of food produced for human consumption is lost or wasted. This equals a total of 1.3. billion tonnes per year, which is a large, unnecessary burden for the environment and the economy. Research and development have delivered a wealth of resources for understanding food waste, yet little is known about where food wasting occurs in the home. The study begins with a literature review of articles that deal with food waste and consumer behaviour, reflecting on their definition of ‘waste’, approach, findings and recommendations. Having noticed a lack of convergence in the literature, and an absence of research into digital technologies for the study of food waste, the potential for incorporating novel technology probe methodologies is explored. Building on the proliferation of internet of things devices, the ‘smart bin’ is introduced as an effective intervention for making visible routine household food wasting practices. These data were then triangulated with user interviews, leading to an enriched qualitative discussion and revealing drivers and mitigators of waste. This paper concludes with some reflections on the smart bin as a domestic product and how it might synthesise previous understandings of consumer behaviour, leading to better informed food waste policies and initiatives.

## 1. Introduction

Mounting concerns over climate change and a move toward greater environmental sustainability bring food consumption and dietary habits to the fore. Globally, nearly one third of food produced for human consumption is lost or wasted. This equals a total of 1.3. billion tonnes per year. It has been shown that if Europeans could reduce waste in the home, the point at which the environmental cost is the highest, effects on climate change could be reduced [1].

Research into the main drivers of food waste and potential strategies for limiting it has grown considerably. A 2018 systematic review of food waste literature found that academic output had doubled in the preceding five years [1]. This was one of seven similar reviews published in the same period [1,2,3,4,5,6,7]. There is clearly a significant effort being conducted to gather and consolidate the existing research. This is no surprise considering the wealth of understanding on how to deal with food waste, both in research and in practice.

The aim of this paper is two-fold; first, to present a gap in the knowledge that emerges between different strands of food waste research; second, in light of this literature, to demonstrate the value of using a technology probe, a novel methodology drawn from the computer sciences, for understanding consumer behaviour.

Beginning in the conceptual grounding and translating into findings and interventions, the research setting or ‘where the responsibility lies’ with food waste is a contentious topic [7,8,9,10]. It can be broadly drawn between two perspectives. One suggests that waste is an individual action rooted in human agency, or the capacity for people to make conscious, rational decisions about their waste. The other argues that wasting practices are a product of their social and economic context, and are unique to different peoples, localities and resources.

These perspectives are divergent, reflecting opposing socio-political positions in the food waste debate. Assumptions regarding the key drivers of food waste are conducive to the context and setting with which researchers approach consumer behaviour, and ultimately what their findings indicate. Advocates of consumer agency suggest that personal testimony is a sufficient source of evidence, collecting data predominantly through quantitative surveys. However, these are not necessarily representative; data are selectively reported, there are few large-scale data rich studies and they are difficult to administer. Their findings largely prescribe incentives at the front end, encouraging responsibility in purchasing and planning [11,12,13].

Critics suggest these actions do not fully engage with embedded practices. Instead, they base their findings on the long-term study of human–food interactions and suggest a widespread transformation in practices of production, retail and consumption [9]. Due to a series of biases and constraints, they fail to appreciate the full picture of household food waste or produce generalisable insights. This reveals that there is little useful knowledge about where food wasting occurs in the home, and few useful mitigation strategies beyond education and purchasing campaigns.

The technology probe is a research intervention that studies how new artefacts fit into the everyday lives of users. As such, it may speak to a future in which food waste issues are resolved through smart products and devices, whilst generating insight into consumer behaviour now. Purchasing patterns [14], dietary habits [15] and food waste [16] are already increasingly interfaced through digital technologies and subject to systematic, highly granular data collection at scale. These data are considered to have significant social and economic value, generating individualised insight for users to encourage certain practices, and allowing public and private organisations to strategise based on aggregated patterns in behaviour. There is a clear trajectory toward generating more of these devices; however, there is a deficit of research into applications and limitations [17].

By ‘probing’ food waste practices, we might begin to learn what this future might look like, whilst pre-empting potential design considerations. For example, in Hutchinson et al.’s [18] defining paper, the authors consider digital information communication technologies for maintaining intimate relationships between family members. They were able to reveal the needs and desires of groups of users, to consider real-life scenarios and use cases, and engage participants in devising new ways of designing technology [18,19].

Significantly, this led to considerations regarding how users both adapt their behaviour to technologies and remake technologies according to their own values [18]. Smart kitchen appliances are gaining considerable attention as a more effective instigator of behaviour change, particularly around issues of sustainability, nudging consumers into better practice [20]. Their impacts, however, frequently fall short due to lack of understanding of users, their context and the complexity of problem they seek to resolve [17,21]. This paper introduces the ‘smart bin’, a device that records food waste to learn about consumer behaviour, filling the gap between different strands in research whilst inspiring future design.

As such, this is not a proof of concept or prototype but a demonstration of the technology probe methodology through an investigation into food waste. As summarised in Figure 1, the paper begins with an exploration into the accepted definition of waste within academic works, followed by a review of some of the key debates arising from understandings of consumer–food interactions and how these translate into interventions. Next, the technology probe and some of the potential gains to be made are introduced, touching on each of the core questions involved in this approach. The final section reflects on the smart bin and how this has made visible routine household practices, leading to amplified qualitative discussion.

## 2. Defining Food Waste

Historically, in urban contexts, public waste management was focused on removing potentially harmful substances or materials away from human settlements. Waste was understood as a *‘necessary social evil’* to be disposed of and hidden away. Food waste has been attributed to a shift in line with the post-war period of affluence, where access to resources became relatively abundant and frugality toward food was no longer considered a social virtue. A growing body of work, however, recognises waste both as a key indicator in defining the current socio-economic context and for its central role in environmental and cultural politics [10].

These academic works have approached the conceptualisation of ‘food waste’ in several ways. A clear cut example would be Papargyropoulou [22] (p.108), who states that ‘*food waste, or losses, refer to the decrease in edible food mass throughout the human Food supply chain*.’ A more human or nuanced perspective, however, would account for the subjectivity in determining how food would become waste, particularly useful in a domestic context. Pongrácz and Pohjola [23] (p.142) argue that *‘the label ‘waste’ does not necessarily mean that the thing is an ultimate waste, rather, it means that it will be treated as waste.*’ This assertion makes space for the emergence of sustainable resource management, grounded on the notion that ‘waste’ can be a ‘resource’. What sets food waste apart from other forms is that it is inherently biodegradable and, as such, can be repurposed as either fertiliser for plants or to produce energy [23].

However, defining waste is a key challenge and most studies in practice tend to draw toward what is culturally relevant to the context and can be normatively defined as ‘avoidable’ or ‘unavoidable’. The Waste and Resources Action Programme (WRAP), a UK based charity founded in 2000, has proven extremely influential in this. Their definition [24] accounts for flexibility while situating waste inside the home and is as follows:

Avoidable waste includes food and drink thrown away that was, at some point prior to disposal, edible (e.g., slice of bread, apples, meat).

Possibly avoidable waste includes food and drink that some people eat and others do not (e.g., bread crusts), or that can be eaten when a food is prepared in one way but not in another (e.g., potato skins).

Unavoidable waste is waste that arises from food or drink preparation that is not, and has not been, edible under normal circumstances (e.g., meat bones, egg shells, pineapple skin, tea bags).

It is suggested that up to 60% of UK food waste is deemed avoidable [8]. Re-purposing as compost, however, is often considered a method to offset food waste [25], with some papers claiming it is central in defining a conscious consumer [26]. Further claims can be made about feeding leftovers to animals [27,28]. This is ultimately reflected in the ‘responsibility’ debate; while composting is a more ‘conscious’ practice than simply binning food, does this assuage or undermine the motivation for conserving food in the first place? 

Evans [29] attempts to re-locate the definition altogether, rejecting the understanding of waste as a separate social category and more as an embedded form of everyday life. This moves beyond an orientation based on recycling, to achieve a reduction in the sources of waste. In this instance, Evans deviates from the purview of the home as a black box, equating the amount of food entering and leaving a household to minimise waste with little to no understanding of what occurs inside. Alternatively, he joins a host of authors who view the problem in terms of redefining consumption practices as a broader socio-cultural discussion involving infrastructure, governments, NGOs and industry [30,31,32,33].

## 3. Understanding Waste and Consumer Behaviour

The empirical study of food waste, as with the accepted definition, is varied and convoluted. Differing streams yield distinct results, further intensifying the conceptual debate. In this section, the following key contexts that manifest human behaviour about food waste are discussed, as emerging from the literature: (1) personal testimony; (2) human-food interaction; (3) societal discourse. 

### 3.1. Personal Testimony

#### 3.1.1. Description

Personal testimony is the participant’s account of their food wasting practices. Data are most frequently generated through quantitative questionnaires and diary studies in which a pre-defined group of people respond to a list of standardised questions, either in a single intervention or over a set period. In other cases, interviews with participants allow a richer discussion, with greater flexibility toward question and answer. Quantitative studies are preferred overall as they can easily access a large sample and draw insights across populations; they tend to explore varying social-demographics and, while there is a concentration of studies in developed countries, there is a range of studies located across Europe [34,35], Australasia [25,26] and North America [3]. Three exceptions identified for this review were in Uruguay [12], Qatar [36] and Egypt [37]. Some studies focus solely on one demographic, be it the youth [13,35] or compare across them [35,38]. It appears that cultural differences are significant; Spanish and Italians being much more proactive in their adjudication of food edibility than British consumers [35].

#### 3.1.2. Findings

While the depth of understanding may be limited in specific cases, for quantitative surveys, it is possible to apply statistical regression [39], causal maps [40], principal component analysis [41] among other original attempts at modelling consumer behaviour [42]. This is evident in the similarities made in drivers of waste in different studies. Eating outside of the home [35], increased income and expenditure are widely associated with increased waste [40]. Factors of affluence [43,44], time-management [27], convenience culture and consumer price-quality relations [12] reveal much about the value consumers have for their food. This is intensified by a lack of understanding and awareness on the environmental impacts of their waste [25,40] and limited efforts made to aid in planning, particularly the understanding of ‘use-by’ dates [28,35,45]

Interviews can reveal more affective and contextually driven notions of food waste. Moral norms are not found to be influential [46]. Similarly, ideologies tend to be inconsistent with practices including vegetarians, vegans and those that consider themselves green consumers [25]. For some, participants are aware of the need to do more but are also careless [26,40,41]. In other cases, there is a complete lack of recognition of responsibility [47].

From this understanding of food waste, key recommendations for mitigation focus on motivating consumers to be more responsible [36,42,48]. Aschemann-Witzel et al. [6] frame food waste as a problem of inertia and disregard among consumers, with education and efforts to change attitudes considered as key areas for future interventions [34]. In particular, guilt is a frequently reported driver of avoiding food waste. Septianto et al. [49] have shown how this can be harnessed, emphasising the importance of emotion in media campaigning; embedding consumers with gratitude to instil a greater sense of value. Wikström et al. [50] stress the importance of doing this through appropriate packaging.

#### 3.1.3. Implications and Limitations

In terms of amounts of food wasted, studies that attempt to track patterns of consumption must rely on self-reporting and this is widely stated as a limitation to their validity [12,25,35,40,43,47,48,51]. Many of the surveys are undertaken online, which serves a self-selection bias [12,35,40,49]. Further difficulties include the lack of perceived consequences; in other similar campaigns for environmental action, such as recycling or reduced energy consumption, there is often a direct and obvious impact on the participant (cost of energy) [42]. These factors combined can potentially lead to unreliable testimony and either passivity or misrecognition among participants.

### 3.2. Human–Food Interaction

#### 3.2.1. Description

While personal testimony appears to be the most prolific source of data for consumer behaviour, likely as they are perceived to be the most practical, generalisable and easily translated into policy, human–food interaction studies are equally valuable. Predominantly, these studies involve researcher observation of participants engaging with their food, generally in the home and in combination with supporting interviews. While they take a smaller sample size, they can record richer data, with greater depth.

#### 3.2.2. Findings

Evans [29] completed 8 months of ethnographic observation and interviews with 19 households in Manchester, UK. His analysis reveals the different ways consumers’ eating habits determine what happens to the food before it is even considered waste; how ‘excess’ or ‘surplus’ seamlessly becomes ‘waste’. Despite growing awareness and infrastructure, many groups and individuals remain disengaged with the systems of food recovery, including hygiene practices and lack of trust.

Lazekk [52] spent 4 months observing and intervening in a group of students. His conclusions included the necessity for such a methodology to realise the underlying social practices that result in food waste mitigation, particularly those surrounding the sharing of food as a mechanism for counteracting over provision. Furthermore, the lack of awareness participants had in their own actions undermined the previous assumptions about personal motivations to prevent waste.

Several authors state identity as a key driver of food waste [53,54,55]. Identity is understood as a an organising principle by which actors can be socially recognisable, while shaping the way they act in accordance with the world around them [56]. Open ended observation and interviews can draw out aspects such as a self-perceived and or culturally defined need to be a ‘good-provider’, resulting in excessive purchasing and waste. It was consistently found that households would provide surplus food routinely, which would over time lead to large amounts of waste [29].

Examples of design considerations for food include the work of Ganglbauer et al. [20] and Bucci et al. [57], who explore the use of smart fridges. They combined observation of participants and user input directly in the construction and testing of technologies. Their findings supported the disparity between aspiring food practices and actual recorded data [4]. The evidence pointed toward smart fridges as effective strategies for food waste prevention, although few large scale studies have been undertaken [5].

Morone et al. [58] diverge from the conventional methods of research, developing an experimental design in which a group of students were asked to purchase, cook and consume food as a collective. The experiment lasted a week, in which they were questioned, observed and their waste was recorded. While there was no clear advantage over the control group, households that practiced food sharing would often include key ‘enabler’ factors (skills, environmentally friendly and collaborative behaviours) that would provide a positive influence on participants. This speaks to both the importance of individual knowledge and shared practices on food preparation and storage.

Kim et al. [59] have pioneered a co-design approach to media campaigning, transcending the expert-led dynamic of previous initiatives. This was achieved first through focus groups, then a campaign and, finally, a ‘fridge audit’ to triangulate and quantitively confirm the impact of their efforts. Their combined analysis reveals the value in opening up the campaign narrative to reflexivity among consumers; allowing a genuine form of agency to flourish, rather than attempting to shape the best, most economically sensical decision in retail [60].

#### 3.2.3. Limitations

As is the case with personal testimonies, human–food studies have problems with participant bias; there is frequently a lack of trust as participants desire to present themselves in a good light in front of the researcher [53]. It is difficult to account for the actual mass of waste as record taking interrupts the routines that researchers are attempting to observe [61,62]. Ethnographic studies rely on a researcher’s interpretation of important data, are difficult to generalise, costly, time consuming and, in comparison to online surveys, demand significant cooperation from participants [30].

### 3.3. Societal Discourse

#### 3.3.1. Description

Human understandings and behaviour regarding food waste are revealed through discourse in the public domain. This ranges from the use of language, narrative and or efficacy of existing food waste campaigns to consumer discussions on social media. These research settings reflect how society engages and understands food waste at different levels, how consumers respond to interventions and how they discuss food waste in their personal circles. Data are scrapped from internet sites, collected through food labelling or advertising campaigns.

#### 3.3.2. Findings

Aschemann-Witzel et al. [61], emerging from quantitative studies on consumer behaviour, acknowledge several key indicators of success in campaigns that increase awareness and motivating consumers, including capacity building, redistribution channels and supply chain opportunities. Närvänen et al. [63] found that by promoting ‘positive messages’, specifically those related to creativity, aesthetics and ethics of food waste, campaigns were able to embed issues of sustainability, overcome individual responsibilisation and challenge socio-cultural issues of consumption. Similarly, Specht and Buck [64] identify spaces within Twitter for discussing food waste and explicate how activism and membership of likeminded groups is a good indication of motivation for mitigation, whilst also a wealthy source of information on how consumers understand their waste.

#### 3.3.3. Limitations

In each case, the authors recognise that this is based on discourse alone and has no means of measuring the impact of such campaigns or discussions in practice [65]. The authors identified many limitations, including those related to the unique cultural context of interventions, the lack of depth in studying participants lives, the assumption that discussions and initiatives were influential and the difficulty to generalise, and that discussion on social media tends to be self-aggrandising rather than an accurate picture of food waste patterns. Thus, while societal discourse is a significant means for understanding the social reaction of people in public spaces, it does not speak to what they actually do in practice.

### 3.4. Reflections from the Literature

A key problem of food waste testimonials is that they rely on the objectivity of the consumer. Study biases are reflected in their conclusions; volunteers who broadly underestimate or miscalculate their amount of food waste are also likely to underestimate their own role in food waste, to displace blame away from themselves, and on to supermarket offers [59,66]. Considering these biases, most quantitative studies situate their analysis in the place of purchase. They anticipate that food will be wasted before it has even left the supermarket. Therefore, waste is often treated as if it is a choice, based on sound economic decision making. From studying human–food interactions, scholars suggest this is overly simplistic and, in effect, ‘blaming the consumer’ for the environmental burden associated with food waste [9].

With regard to human–food interaction, scholars suggest that eating practices are related to a complex set of irrational behaviours and drivers [23]. Conscious actions that lead to waste reduction are ‘seldom socially oriented, seldom exposed to peer pressure and very reliant on purely ‘altruistic’ attitudes’ [67]. The most significant mitigating factors are heterogenous and implicit within habitual activities. The motivations for such are not necessarily associated with waste prevention and, therefore, are not easily identified by participants without extended observation [68]. The future of food waste prevention must, therefore, address ‘a web of interlinked practices making up the everyday life activities, infrastructures and meanings of consumers’ [30].

Sociologists Delormier et al. [31] take inspiration from Anthony Gidden’s structuration theory; considering how consumption practices are an interplay between human agency and social structure. Consumption is understood according to ‘the social and material contexts through which practices are ordered and (re)produced’ [9] (p. 430). The potential for such an approach is echoed in studies that deal with both group practices and personal testimonies [52,58]. This sociological perspective could potentially allow research to analyse the interplay between these rival understandings and, combined with the correct methodological tools, could vastly further the understanding of a notoriously ‘invisible’ social phenomenon [69].

It appears that there is a willingness among actors to reduce their environmental impact, including through the mitigation of food waste; however, the question becomes the following: what can initiatives do beyond educating consumers and shifting purchasing practices alone? Efforts of supermarket chains and media campaigns have often proven to be insufficient for lasting change and, in certain cases, to have ulterior motivations. Previous studies have indicated that embedded routines are paramount and that while consumer motivations exist, they are rarely effective and are often confused and or contradictory. By incorporating the desire to do more into everyday practice, routine and even culture, how can action and responsibility of wasting food be re-positioned to where it occurs, in the home? Following this, hopefully some of the confusion surrounding food waste can be alleviated and consumer behaviour identified as part of a broader environmental-systemic change, rather than the driving force behind the culture of waste [7,29,31,32,70].

## 4. Technology Probes

The technology probe has emerged from human computer interaction (HCI) studies, a disciplinary intersection between engineering, social and computer sciences, which rose to prominence along with personal computing in the 1980s. The approach was first popularised by Hutchinson et al. [18] (p.18). They define a probe as ‘*an instrument that is deployed to find out about the unknown-to hopefully return with useful or interesting data*’. Inspired by Gaver et al. [71]’s cultural probe, they take a situationist approach to research, provoking a reaction from their participants. The technology probe is intended to extend this through longitudinal data collection, an extended intervention in the day to day lives of users. As a methodology, this involves research participants in the process of formulating and developing a potential product that serves a purpose for their use, while simultaneously leading to novel findings. From its origins, this is an interdisciplinary means for data collection, attempting to answer a real-world question in a real-world setting.

The societal context in which the technology probe is embedded must be defined and understood, with clear relevance to the question at hand. The probe must be simplistic in design, with as few functions as possible and a high degree of usability. It must be engineered such that the functionality is smooth in the field and that it serves a purpose to the user. It is also fundamental, however, that the study remains open-ended, reflexive and adaptive. It is often encouraged that participants engage with the probe as they see fit, fostering creativity and leading to insights for new technology design. To maintain the marriage of quantitative probe data with qualitative user/designer input, the study must be fluid to developments over time.

Since their inception, technology probes have been used to tackle interfaces between societal demands, individuals and designers. Technology probes reject the assumption that interventions must gather only unbiased ethnographic data; by serving a purpose to the user, it must alter the context in which it is used. Technology probes are, thus, especially useful for learning about changes in human behaviour, or how to develop a technology that yields favourable behaviours. Edwards, McDonald and Zhao [72] conceived a probe that encouraged healthy activity and exercise among teenagers. Through collecting quantitative data counting steps and calories, combined with a series of workshops generating rich qualitative insights, it was found that the device not only led to increased exercise among adopters, but also gave users a feeling of empowerment and control over their physical health. The probes they designed have since become ubiquitous among consumers.

More recently, technology probes along with commercial devices are becoming key tools to overcome personal inertia and enforce a stricter regime of so-called sustainable behaviour. For domestic consumers, the leading strategy for ‘doing one’s bit’ is a practice of self-regulation and social pressure. Smart-home tech and appliances are increasingly becoming part of everyday lives, capturing, manifesting and reporting data in order to help people make sense of their behaviour. There is some contention, however, as to whether this pathway is the most effective in generating a greener future [9,21,73,74,75]. First, by focusing on the individual rather than the collective, the responsibility for societal, government and corporate action is undermined. The potential for significant and lasting change, and the greatest portion of greenhouse gas omission, is currently performed at a government policy level [9,75].

Second, individual contribution is often divisive across the social demographic. Users are considered wholly rational, economically driven subjects [73]. In such cases, practices of consumption considered ‘ethical’ or ‘sustainable’ can be inaccessible to those excluded due to issues of class, race or gender, and often act in aversion to existing cultural traditions [21,75]. As Kwon [76] reveals, data recorded via a ‘shower probe’ can reveal intimate, personal information and lead to a more fruitful discussion on water consumption. Technology probes can lead to a deeper human connection to both the problem at hand and other participants. The operationalisation of intimate data, therefore, can potentially be extrapolated across broader swathes of society, assuage group divisions and foster political mobilisation for collective change [73].

Following Delormier [31], this study moves away from the analysis of consumer behaviour in individual and isolated cases, toward treating food consumption and waste as a social practice. The technology probe is proposed as an ideal methodology for both better understanding and indicating appropriate strategies for the problem of food waste that build on the smart-home, recording quantitative waste data, making visible wasting practices and situating them within a social context through rich discussion and participant input. 

## 5. Methodology

The design of the smart bin is shown in Figure 2. It is a compost caddy equipped with a set of weighing scales and a camera, attached to a Raspberry Pi computer. Images and weight data are recorded of participant food waste, taking a measurement every time the bin lid is opened. The aim was to collect data about consumers in the home, to learn about food waste and wasting behaviours, and to understand how users interact with a smart compost bin to uncover the ‘invisible’ aspects of food waste. Meanwhile, inspired by the growing prevalence of smart devices for data collection and user reflection, the smart bin was evaluated as a domestic product, identifying potential improvements. Finally, this was used as inspiration to consider future use-cases, design opportunities and experiments to be undertaken.

The bin was designed with the aim of connecting domestic consumers with researchers and other stakeholders with a vested interest in food waste data, be it consumers themselves, local municipalities or food technologists. Thus, both the usability of the probe and the usefulness of the data collected in the experiment were considered. It is important to note that the purpose of the probe is not to make conclusions about the artefact of food waste based on a representative population, but to motivate this study as the first step in formulating a future appliance. It is the type and variety of data that the bins collects and how this speaks to human behaviour which is of interest, rather than ‘what’ food waste is placed in the bin. Thus, the smart bin was evaluated with the participants most available to the first author during the time of study, a snowball sampling of households in the county of Dorset, UK. This was effective in showing that even in a relatively homogenous and accessible sample, the richness and breadth of data collected is significant and, considering that the point of analysis is the interaction with the probe, is a sufficient group to reflect upon its culture of use [77].

Participants for the experiment were those in the sample area that prepare and consume food at home, which must then dispose of waste material, and perhaps had a compost bin of their own or, as is now common throughout the UK, a biodegradable waste collection service that demands they separate food waste from other waste. Therefore, the catchment for potential participants was extremely broad, with most households fitting this criteria, and that participation in the initial phase of the experiment was relatively passive, as they simply replaced their existing caddy with the smart bin. This preliminary study included 10 households and 19 individuals were interviewed between April and December 2020. A full description of participant households is given in Table 1.

Participants were given the bin for a period of two weeks. The first author then remotely transferred and manually coded the pictures and weights, importing information into Excel. Using R data processing software, an html was generated to be shared with the participants to give feedback on their performance in the experiment. This was designed to envisage what a digital interface for a future commercial version of the smart bin might look like. This then became the basis for an interview in which households were asked to describe their consumption patterns, trying to understand how this reflected their interaction with the probe.

Household members participated in the interview voluntarily and not all chose to do so. Interviews lasted between 30 and 60 min. During the first half, the participant reflected on their week of observation. Participants had been asked to interact with the bin as they normally would do their compost caddy. Participants were, thus, asked to reflect on this, any notable experiences or feelings. The key differences reported were that the bin required a power source, so in some cases, it had to be placed in a different position as their usual bin, and that participants had to use biodegradable plastic waste bags, which some did not like.

The feedback html was given to the participant in advance of the interview. It included the accepted definition of waste (avoidable and unavoidable) and three visualisations; the first tracking consumption across time, the second showing the most frequently wasted products and the third revealing the financial and carbon cost of avoidable food waste. The figures were interactive so the participants could hover their cursor over each point to view their wasted items. When observing these graphs, participants were asked to ‘fill in the gaps’, giving context and to describe how reflecting on the data made them feel. This half of the interview was participant led; individuals were frequently keen to challenge the results, explain away their guilt or to explore their own habits in the context of others.

The second half was made up of semi-structured questions, designed with insights from the literature review and the following key areas identified by Schanes et al. [1]: planning, shopping, storing, cooking, eating, and managing leftovers. These questions were aimed at assisting the participant in explaining and understanding their habits. While the previous section was emotionally provocative, these questions generated greater rationale in explaining contexts, leading participants to discuss their culture, familial history, and future ambitions.

Transcripts of these interviews were coded using NVivo. A thematic analysis was conducted to reduce the dimensionality of the data [78]. Following Boyatzis [79] (p. 161), a broad range of codes were identified “that at minimum describes and organises the possible observations and at maximum interprets aspects of the phenomenon”. The coding pursued a hybrid approach, first employing inductive reasoning and developing data driven codes, then using existing theory, aggregating key umbrella codes; identities, emotions, practices, contexts, social habitus and knowledges [80]. This methodology allowed for a clear comparison between the raw qualitative data, literary insights and evidence provided by the smart bin.

## 6. Results

The results are broken down into quantitative findings, evidencing the potential insight to be gained from the raw data produced with the bin, and then the themes arising from the interviews, using the bin data as an instrument for enriched discussion. The quantitative data is intended to demonstrate the applicability of the bin as a tool for data collection, in particular the breadth and granularity of information. The qualitative section first reveals the bin as a means for personal reflection and how the participants react to their behaviour being recorded; and second, as a situated device influencing peoples’ practices and leading to new strategies on how to prevent waste.

### 6.1. Quantitative; Making Food Waste Visible

Here, some of the most confounding and poorly understood questions within food waste literature were considered. The sections above have evidenced that a wide number of cases have been identified for manifesting people’s behaviour about food waste, yet little can be reliably said about what goes on inside the home. The bin presents a series of novel findings that speak to the root of these questions, making visible the action of food waste accurately and without significant bias.

#### 6.1.1. Waste Variability Inter and Intra Households

The bin demonstrated a rich variation in wasting patterns between households and across time. The quantitative results and sample description are summarised in Table 1. It shows the demographic of adults and children, the total number of times the bin was visited during the study period and the number of different items deposited. Each has a broad range, and the number of bin instances would appear to increase with more residents. As might be expected, it is indicated that more people will waste more things.

Next is the proportion of avoidable and unknown items. Avoidable items are defined according to the wrap definition above. The proportion of avoidable items wasted ranges quite significantly from 7 to 40%. There is no obvious connection between this and household size here, suggesting that perhaps while total waste could be linked to household size, avoidable waste may be down to individual behaviours or other more complex dynamics working within the household.

Metrics according to weight support this; total weight ranges from less than 300 g to almost 7000 g across households, consistent with household size. Proportion of avoidable waste weight ranges from 17 to 67%, with the average proportion of total weight being 29%. In households 6–10, avoidable waste is a significant proportion of their total waste, whereas in 1–5, it is less so. The latter groups are overall larger households while the prior are smaller, with the exception of household 7, which has young children. This would suggest that while larger households produce more waste overall, they are more efficient in minimising avoidable waste.

The cost of food emphasises this point, revealing a further disjuncture between items, weights and individuals. The UK national average of avoidable food waste per week as of 2018 was 1330 g at GBP 4.04 per capita and 3170 g at GBP 9.62 per household [81]. However, this sample is considerably lower than their estimates according to weight, with costs ranging from GBP 2.80 to GBP 14.26; 6 and 9 produce significantly greater costs. Initial patterns indicate that while the single household with children has significantly more avoidable waste, the financial value of that waste is much lower than in the homes of just two adults. Equally, larger households, with greater amounts of total waste, proportionately produce much lower waste in terms of monetary value. 

The final column shows unknown items, which do not necessarily speak to the variety in the data; however, they do show the limits of the bin and a reflection on the reliability of data collection in each household; the lower the percentage, the greater the certainty. These occurred when images were unidentifiable by the author, when the bin was too full, condensation formed on the lens due to hot items, such as tea bags, the light on the camera stopped functioning or if the bin was not plugged in, as observed in the selection of photos at the top of Figure 2. Other unreliable features include maintaining the accuracy of scales and tracking their calibration over time. There are significant points to be made regarding consumer behaviour between households according to these data; however, future experiments could benefit from a longer period of observation, accounting for ‘one-off’ expensive waste items that skew the results, and a broader sample size, including more houses with children.

Having shown the aggregated data, Figure 3 demonstrates the full spread and detail of the data for a single household throughout the experiment. The graph in the centre is an example of what each household would receive in their feedback sheet. The x axis shows time, and y axis is weight of waste in the bin. Going from left to right, the blue line increases as the participants add items to the bin. Black dots and red crosses indicate instances of unavoidable and avoidable waste.

Among the avoidable items were mushrooms, bread and oranges, as shown in the pictures below. The pictures are examples of the raw data; however, participants were only provided with the item description, time stamp and weight. The left and right images show ideal cases where items are introduced to an empty bin, whereas the central bread image shows how items may become mixed and difficult to identify. While the bin does not reveal who is exactly responsible for the waste, it does show when, what and how much was wasted.

Given this information, patterns can begin to be identified in the data. Each peak indicates the bin slowly being filled by participants, followed by sharp troughs as the bin is emptied and the weight returns to zero. The first peak climbs to 1000 g of waste over more than a day and is then emptied. Immediately after, in stark contrast, the bin is filled to nearly double that weight and emptied again in only a few hours. The bin then resumes a steady increase in weight, reaching a larger peak weight with fewer avoidable items. The red highlighted sections indicate a discrepancy between the two apparent patterns emerging from the data including shorter, more frequently emptied bins, with greater avoidable items and less frequent, heavier bin loads with fewer avoidable items. Figure 3 does not contain error bars, nor does it attempt to compare or draw averages between the households due to the limited size of the sample. It does, however, reveal intricate arrangements that can be extrapolated to make conclusions both for researcher and user; for example, as was common across the households, the bin was not empty for long, and was seemingly only emptied when there was a need to make space for more waste.

These patterns are further explored in Figure 4, selecting households 1, 4 and 9 to reveal the observable differences and similarities between them. Each household routine is unique, and their regularity reflects their configuration. Household 1 has five persons, resulting in quick filling and emptying of the bin, five times over two weeks. Household 4 with two persons fills the bin three times, at a much lower amount but with similarly uniform peaks and troughs. Household 9 seems to use the bin regularly for one week, then sporadically around that. The bin is filled and emptied systematically, at similar weights and intervals. Avoidable items, however, are irregular, sometimes together and in small amounts; at others, infrequent and large, irrespective of other factors. Intra and inter household variation in wasting is significant, revealing a consistency in overall weight but disorder among avoidable items. While this speaks to the granularity of household practices and human behaviour, as of yet, little can be said about mitigating avoidable waste, not revealing any specific rationale with the data.

#### 6.1.2. What Households Waste

The bin reveals a range of waste products; however, the sample is significantly skewed toward a small number of them. Figure 5 shows the accumulated weights of all the items across the sample. The most wasted items by weight were tea bags, followed by coffee and leftovers. This is unsurprising as tea and coffee necessarily produce waste, and the by-products are rarely consumed in the current sample. Their waste produced is presumed to be directly related to the amount of tea or coffee consumed, unlike leftovers or other avoidable wastes that can be more easily assigned to human mismanagement. It is expected that these items may also be biased by the sample context, being rural UK households, and that these items may be different in a different area. Equally, the level of balance in diet may be greater or lesser depending on cultural or ecological context. Tea and coffee are also frequently disposed of directly from a liquid, having been brewed in hot water, which adds considerable weight to the product.

Again, there is little consistency in avoidable items. The majority of weight from certain items may be attributed to certain households. Figure 6 disaggregates item weight by household and indicates which were avoidable or unavoidable. Each household would have observed a similar visualisation as part of their feedback sheet. Households 6, 7 and 8 have leftovers, oranges and bananas as their topmost wasted items, and in each case, a significant portion of their top items were avoidable. Household 7 is the only house in the sample with young children, confirming the struggles to mitigate waste noted by parents in previous literature. Equally, households 1, 2 and 3 seem to account for most of the tea and coffee. Excluding household 8, these had the greatest number of residents. These households were included here to demonstrate the diversity between them and the items used, whereas households 4, 5, 9 and 10 were even more highly skewed. These graphs have evidenced the range of products in the sample, revealing a high level of bias that might be better addressed in future studies by approaching different geographies, climates and social groups, which might vary in products accordingly.

#### 6.1.3. Amount of Household Waste

As observed above, houses waste different items in varying amounts; however, there is also a discrepancy in the rate of wasting, with some households wasting little and often, and others infrequently and in large quantities. Figure 7 displays the range and then compares the grams per instance between bin visits across households. The box and whisker plot shows the median, interquartile range and anomalous values for each household for the weight of items. Household 7 is among those with the fewest instances of wasting, yet they are among the greatest in increase in weight. The opposite is true for houses 1, 2 and 3. Household size might be associated to how and when the waste is thrown; for example, houses of 4 to 5 adults seem to be more regular, whereas the range in households of fewer people is much greater. Equally, anomalous values are greater in smaller households, perhaps reflecting a lack of flexibility. In contrast, letters accompanying each box indicate a significant level of difference between certain households, and similarity among others, according to an analysis of variance. Houses 1, 2, 5 and 6 (bc), 3 and 8 (b), can be grouped apart from the others, revealing a similar distribution of grams per instance, and that other patterns can be extrapolated that do not have an obvious connection to size, items or weight. Frequency and weights per bin visits are evidently complex and varied between households.

Equally, items are wasted in different amounts and frequencies. Figure 8 shows the grams per visit disaggregated by item and food group. The single largest bin visit throughout the sample was for leftovers. In combination with Figure 7, this can be attributed to household 9. The next largest bin visit was for oranges and bananas. They also had the largest range. These were largely from household 7. In combination with Figure 5, there is a clear difference between the accumulated and separated weights among the most popular items, with tea bags, vegetable choppings and coffee being deposited little and often, as might be expected. Around half of the items were only wasted once. The least change in weight was unsurprising, coming from very insignificant items, such as sink waste, coriander and cheese rinds. By triangulating these data, outliers and non-normal distributions that could be easily misreported or underrepresented in a questionnaire can be accounted for. While the smart bin has only begun to scratch the surface, it has revealed a high variation in how much households waste of different items, which demands further explanation.

#### 6.1.4. When Households Waste

Households waste things at different times, with little comparability within the sample. There is an observable difference according to the total weight and time of day between households, as shown in Figure 9. Time of day is defined as morning (5:00–12:00), afternoon (12:00–18:00), evening (18:00–21:00) and night (21:00–5:00). These periods were chosen to encapsulate mealtimes and the range of times at which people in the sample would eat, including morning breakfast, afternoon lunch and evening dinner. Excluding perhaps household 3, each household seems to favour one time of day above others for using the bin. While some are quite balanced in this regard (1, 2, 5, 7), others are significantly skewed (6, 8, 9, 10). Overall, there seems to be little coherence between households, with a wide diversity across the sample.

Items are wasted at different, unexpected times, as the same comparison is made between some the most wasted products in Figure 10. These were coffee (spent grounds), leftovers (uneaten, cooked food), tea bags and vegetable choppings (inedible parts of common household vegetables, including onions). Certain items here have a conventional or assumed period of consumption. For example, stimulant drinks, such as tea or coffee, might be expected to be utilised in the morning, yet they are spread across the day, with the majority of tea being wasted after 21:00. Leftovers are mostly in the morning, perhaps suggesting that breakfast is the most wasted meal. Vegetable choppings are mostly at night, corresponding to cooking for an evening meal, perhaps disposed of after eating. In any case, there seems to be little coherence between products, again with a wide diversity across the sample.

### 6.2. Qualitative Analysis; Realising Behaviours

The quantitative results have shown that the bin as a data collection device can overcome many of the barriers experienced in food waste behaviour studies (expense, confirmation bias) and reveal new insights. This section demonstrates how when confronted with quantitative evidence of their food wasting behaviour, certain themes would arise that the participants were not fully aware of, leading to unexpected explanations and reactions.

#### 6.2.1. Routines and Identities

The most prominent themes were routines and identities. These were both unique to certain households or individuals, or generalisable across the sample. Furthermore, these findings reflected much of the preceding qualitative literature. The routines and habits as described by each household included tea bags in the morning, as part of the process of waking up, or unfinished cucumber and the ends of a loaf of bread, discarded as surplus between shopping visits. Queries from the raw data, for example, why coffee and tea, which presumably are consumed more in the morning, were found to be wasted late in the evening, was explained as that is when the ‘big wash’ happens, or when the coffee machine or teapot is emptied after a day of use. Equally, a wide variety of reasons were given for why different households emptied the bin at different times and rates, for example, to pre-empt the bin beginning to omit an odour, some had different perspectives on when it was full, some emptied the bin because they were intending to be away for a night, others forgot and lamented their lack of foresight. Much of the study was undertaken during the summer months, and participants often described this as a factor in why they emptied the bin at certain times.

Figure 3 of household 3 was a particularly good example of this. It was discovered that the three large peaks, highlighted in red, correlated with the absence of one household member known for their vigilant cleanliness. That member had been away during the highlighted times and had returned to cook frequently between them. The bin is emptied with greater frequency and regularity, at lower weights and with more avoidable items, as the member proceeded to clean the fridge of neglected contents. This revelation proved a rich point of discussion and reflection and led to the household member being congratulated on their raising of the standard in household hygiene in the following ways:


*[A] “I’m just chucking out all the… mouldy [stuff]”*

*[B] “A is such a good cleaner”*

*[C] ”Yeah, well-done A.”*

*Household 3*


Not only does one member adopt a particular identity, acknowledged by themselves and the rest of the household as a vigilant cleaner, they are also enabling other members to perform better practices. Whilst their actions are leading to greater waste, through the bin, they are making visible that which would have been forgotten. Furthermore, that same member held the most concrete ideals on the cost of food waste. Other members experienced shame and regret for their waste but did not connect their feelings with the same practical economics.

Other routines emerged, such as those related to sleep; certain individuals explained late night bin visits and particularly tea drinking as a remedy for insomnia. Seasonal routines also were claimed to be determined by the onset of the summer season, not only as diet and tastes change but also a large portion of the sample grew their own food, and thus experienced food gluts and excess waste. Shopping and storage were considered large deciders of waste, with certain houses adopting systems to keep note of what they have and what they need. Routines were equally broken, significantly due to the COVID-19 lockdown, where participants found themselves at home during the week, where they would usually be away at work. This led to extravagant weekday lunch times, cooking and shopping to combat boredom.

Other identities noted in the study include providers (those who overcompensate food for guests), enablers (individuals with an ideological conviction or skillset that would aid and influence others in waste prevention), bin-pickers (those who refuse to waste anything, even eating from the bin), or roles such as that of working parents and children. Some of these identities can be found in the literature [1,32,53,82], while others emerged from this study. Examples include household 1, where the mother claims to experience recurring nightmares about running out of food (“*it’s always Christmas… and all these people are there… and I’m supposed to be cooking for them but I haven’t got the right food*”), household 5, where the sole occupant went to great lengths to preserve vegetables she could not consume alone in one sitting (“*broccoli, which I really love, I put it in water in the fridge but… if I don’t eat it within 2 or 3 days, It probably has no food or nutritional value and it looks like it’s dead*”), and household 2, where the father voluntarily ate out of the bin (“*Dad picks things out of the bin*”).

#### 6.2.2. Discrepancy between Responsibility, Reported Data and Explanation

While the bin data were able to reveal and cause participants to acknowledge routines and identities that they were not necessarily aware of or did not understand the relevance to food waste practices, there were also disagreements and discrepancies between the accounts participants gave and the evidence produced by the bin. Returning to Figure 3 and household 3, member A is explaining away their actions; however, he is also shifting the responsibility from himself as the disposer of waste to the other household members as those who wasted the food in A’s absence. This is demonstrative of the emotional dimension revealed by this graph and the complexity of ‘wasting’ as a social practice. Other households would blame different ‘wasteful’ generations or demographics culpable for the current culture. Across the sample, household members would frequently challenge the results, giving reasons why they were not responsible. Household 3, when challenged as to why they had wasted grapes, responded in the following manner:


*“it must have been rotten… I’m not happy with that. I mean it’s all gone back into the ground basically, so it’s not wasted, it’s compost… Now that is nit-picking.”*

*Household 4*


Despite being presented with the accepted definition of waste, participants would challenge how their waste did not fall under the given categories. As in the quotation, this could be because compost is envisioned as an equally valued use for food as consumption, others include that the food was grown themselves, that insects had eaten the food, that supermarkets had provided poor quality products, poor control of portion sizes or even simply the waste was unavoidable because the item did not suit their preferences. In this example, it shows that the definition of avoidable waste is not singular, that individuals have a different perspective of waste depending on context and access to resources, and that when presented with these data, participants seem to be more prone to stick to their personal convictions rather than take responsibility for their waste. The probe here reveals participants’ practices back to them in a way they did not expect, clearly provoking an emotional reflex that in some cases dislocates the rationality of the issue. In the following quotation, the participant describes how they grew afraid of the bin:


*“the sin bin, yeah, just it made me think oh god on what am I chucking away”*

*Household 7*


There were moments where the bin itself became the agent that defined what was unacceptable waste, causing the user anxiety as to what they could and could not waste. Participants admitted to accidentally throwing waste in the wrong bin, then moving it for the sake of the experiment. Others would joke that they would have put avoidable food waste in another bin to not be criticised. Often participants were aware of when they were disposing something that might be considered improper, despite all of them saying the bin was passive and fit into everyday life as a normal compost bin. The bin data are divisive, provoking both conformism, causing users to change their habits to minimise waste, and dissent, users refusing to believe the definition is representative of their actions. This seems to further elude to the question of ‘where the responsibility lies’ in food waste, highlighting the discrepancy between different understandings; however, it is clear that in the research process, the probe has a direct impact on the user and their experience of waste.

### 6.3. Qualitative; Provoking Change

The purpose of this section is to show how the technology probe as a research instrument and intervention technology, used in combination with a qualitative discussion, can instigate change in users, revealing how they respond to being measured and resultant actions on waste, and indicating possible areas for design.

#### 6.3.1. Reflexivity

Reflexivity, here defined as the examinations of one’s own beliefs, judgements and practices, began to emerge as the discussion wore on. Significantly, this seemed to be encouraged as the data became more tangible and contextual framing. As Figure 9 and Figure 10 begin to aggregate waste and formulate a perspective, it became more difficult for participants to dispute the evidence, as was provoked by the singular instances of waste, considered in isolation, as shown in Figure 3.

Individuals would focus on the definition, taking the discussion forward. Whilst in many cases the question of ‘waste as want’ might be considered a choice, some participants were aware of their situation, residing in a privileged area of the UK without having to worry about where the next meal comes from. Participants would empathise with those less fortunate than themselves, sometimes in extreme circumstances, and consider how their casual waste might be perceived by someone in poverty. This was particularly prominent in mixed heritage households, where different experiences led to different understandings on how to determine and avoid waste. For example, household 4 were of Malaysian and British descent. They were able to draw on the different stories they heard or experienced growing up and empathise with each other, synthesising a perspective on waste cognizant of a much broader understanding of privilege and access than other households.


*“[coming from Malaysia…] I am driven by the idea of not to waste… food like everything else in the world is finite. And the less you waste… the more there is available to everyone. we hope.”*

*Household 4*


These reflexive moments were embedded with emotional experiences, contextualising the more legible data points of financial and environmental cost with how it made them feel, prompting them to explain their actions and motivations further.


*“Yeah. It just feels so wrong… it just feels that it’s a kind of reflection of your lack of organization”*

*Household 3*


The quotation shows a strong reaction from the participant; despite their waste being minimal in comparison to the UK average, the presentation of the data is influential. In comparison to the earlier visualisations, this table puts the waste in less abstract terms, causing critical self-reflection. The process of the interview seemed to draw participants in, pausing at each stage to reflect. 


*“I felt the shame when I put the [oranges] in… I was like these have sat in there no one is eating them. What can I do… Think of the air miles.”*

*Household 3*


In the quotation from Household 3, the individual explains their experience of wasting, their confusion about how to prevent or repurpose the waste and their awareness of the environmental repercussions. This stage of the interview began to reveal both a connection between individuals, their actions and the impact, and a dependency between household members to coordinate sustainable practice; most participants were no longer rejecting the accusation of waste but searching for potential means to change their habits. This was significant as certain items were wasted multiple times, as household 8 reflects in the following manner:


*“I’m not buying any more oranges. It’s not me. It’s the kids. Honestly… They’re like, ‘Oh, I really want these’… And so I buy them. And then they don’t get eaten because on Monday nights they have clubs, they have after school activities on Tuesday nights. They have after school activities on Thursday night… so they just genuinely aren’t home to eat them. And then by the weekend, I’m so annoyed with the fact that they’ve been sitting here I chuck them out… I went through a stage of… blending them in a smoothie… and I’m just pouring smoothies away because I’m like ‘girls smoothies’ and they have two mouthfuls and then they’re like, ‘no’, and I just wasted 40 min prepping smoothies.”*

*Household 8*


In Household 8, a working mother is struggling to balance feeding her children healthy food, the demands of daily life and not generating excessive waste. Having disposed of almost 2 kgs of oranges over three bin visits, she describes how she is aware that this has been a problem for some time, and that attempts to mitigate this have been unfruitful. The quotation reflects the number of considerations and demands that lead to waste and the efforts of one individual doing their best to prevent it. In this case, the technology probe not only makes visible the action of food waste, it also reveals a contention between the routine and identity of motherhood, and the reflexive and emotional agency of the individual.

#### 6.3.2. Contextualise and Strategize

As tensions arose between the participants’ understanding and practice of food waste, it did not necessarily incite despair among the participants, but instead caused them to contextualise why that waste occurred and, sometimes, to strategize as to how to overcome it. The same mother reflected on how her children would team up and distract her so that they could throw away their food without her noticing. In this case, the bin became a useful tool to monitor their consumption (it was a common surprise among households who would consider themselves non-wasters, then the bin would reveal how other household members had let them down). She also shared several methods on how to mitigate her waste, including a record of every food item stored in the fridge and freezer and how her partner and herself share cooking responsibilities.


*“you are dealing with a north London Jewish boy, you do realise that don’t you. We don’t do things like that, the majority of rubbish is put in the black bag.”*

*Household 9*


Others would delve into why the issue did not occur to them before. In Household 9, a man describes how his urban upbringing in a Jewish family had not led him to be conscious of waste. He would later describe how increasing awareness of environmental issues had led him to change his behaviour to a certain degree but even when faced with the data, disposing of 1 kg of cooked leftovers in two bin visits, the amount seemed reasonable. Cultural background and past experiences proved significant for both increased and decreased waste, whether that be one-off, life changing occurrences or long-term education.

The statement also reflects how the culture of waste is understood. The guilt here is not that the food was not eaten but that it was put in a black bag instead of composted. This sentiment was shared by many and reveals how when asked about their food waste, a common understanding of the responsible consumer is not one who attempts to prevent waste, just one that puts the correct waste in the correct bin. This both shows the reach of the existing waste awareness campaigns and their limitations in getting to the root of the problem. Regardless, in some cases and in contrast to the literature, lifestyle choice and ideological conviction were strong indicators of low waste.


*“well its £3.28 I shouldn’t be wasting… I felt completely ashamed… I can hear my parents saying there are people… in India starving … I think when you’re challenged on anything…, it makes you think, again, and probably, you know, a lot more deeply”.*

*Household 5*


In Household 5, the participant lived alone and was a keen environmentalist. Even the minimal waste she generated was enough to drive her to wish to change. She also admits that the bin brought forth feelings of nostalgia and guilt, and that it had made her think more deeply about her waste. Concrete strategies that households mentioned they did or could do more of included managing the fridge, bringing old items to the front and looking in the fridge before shopping. This was devised in household 2 in response to their wasting of fresh broccoli and cucumber. Household 3 would practice incorporating old meals into the following days’ lunches. Several houses mentioned their confidence in ‘the smell test’. Household 8 mentioned preserving foods such as jams and chutneys and planned to do more in future. Household 7, alongside making waste smoothies and systematizing the fridge, was opting for abstinence.


*“I just would stop [buying oranges]… that’s an obscene amount of oranges. Let’s not mess around here. We just won’t buy that many oranges.”*

*Household 7*


Evidently less effective mitigation strategies were shown by household 9, by ordering meals that have just the right ingredients to produce the accompanying recipe, and household 6 planning meals weeks in advance.


*“So we’re even, at the moment… two or three weeks ahead… isn’t it? … we’ve got a chest freezer over there”*

*Household 6*


Despite these confident assertions, these two households wasted a significant amount of leftover food. In both cases, this was their most wasted item, leading both to significantly exceed the national average cost of avoidable food waste. Certain practices to mitigate food waste and routinised consumption are not conducive to a deeper understanding or responsibility; in some cases, these actions lead to ‘routinely’ wasting food. Even when someone says they are reducing waste, this does not mean they actually are. Once again, there is no clear answer for how to move forward with food waste mitigation; however, while the bin revealed tensions, there were instances where the combination of routine, identity, reflexivity and affect would indicate a more holistic approach.

Following the feedback sheet, the participants were asked how they would feel if their data were used as part of a social media application, where users would share and compete in their food waste, in a similar way to a sports app. This was considered a realistic extension of design and a necessary consideration as similar practices are already used in other behavioural change devices. Notably, many already viewed the experiment as a competition within the household and there were two key reactions. The first group feared an intrusive ‘big brother’ technology, preferring their privacy. Others found pride in their food waste, as demonstrated by the following statement:


*“I would never be ashamed that I don’t feed enough veg”*

*Household 1*


While the information in the tables above might be compromising, revealing instances of food waste that did not fit the self-perception participants had as responsible consumers, some suggested they were keen to share their data in the study and via social media, if given the opportunity, as it reflected upon their pride in their diet. This is significant, as in Household 1, the idea of sharing food waste data appeals to the provider identity, outweighing values of frugality and responsible consumption. It shows that human behaviour is not rational, linear or predictable, and planned behaviour does not seem to change it. The smart bin not only connects waste with consumption, but it might also connect sustainable wasting and eating habits with the deciding factors of history, culture and values, demonstrated by the following statement:


*“We eat the skins now because it became trendy, when we were kids people skinned potatoes, but it’s a different world. If I still had Dad’s potatoes, I would peel them and you’d never get the same again, the creaminess of them is just… [exclaims]… incomparable!”*

*Household 1*


The provider individual in Household 1 describes how changing pressures and social trends have influenced her consumption patterns. In her childhood, before consumption practices began to conform to environmentalism, they would always peel potatoes. As history has progressed this has changed; however, the enjoyment and value in food has not. These quotations demonstrate how culinary shifts are culturally determined and how they lead to changes in identity. The smart bin was able to provoke participants to contextualise their food waste data, while revealing their values. These are clearly interlinked and the bin has uncovered contextual information that offers a pathway for operationalising that relationship, and the certain positive values and sentiments associated with the action of wasting. Of course, there are contrarian or antagonistic instances, where values serve as a barrier to these suggestions (participants admitted to not owning a compost bin due to its untidiness); the solution seems to be in each of these constituent parts working in tandem, influencing and transforming each other. That is, identity and routine, reflexivity and agency working with the bin as a supportive technology might be a more fruitful strategy for avoiding waste.


*‘It’s a salutary lesson, I should put it on the wall’*

*Household 5*


Beyond strategies for mitigating waste, the participants began to think creatively about use cases and alternative opportunities for design. The quotation shows how the user found value in the data collected, particularly the visual component, and that they feel they should be constantly reminded of their waste by projecting it on their wall. Other suggestions included receiving different photos each day, making connections with different practices more apparent, and fostering a stronger affective engagement through the food they had most recently enjoyed. Some maintained diaries voluntarily, asked for buttons or a voice function to maintain a consistent qualitative engagement, where users could describe their food and reason why it was being thrown away as the experiment progressed, rather than feel judgement from the device at the end. This fed into how users reflected on the purpose of their data; one participant asserted that if they had paid for the bin as something to collect data for their own benefit, their attitude might be different; however, their judgement by human researchers was not considered to have the same value. Thus, the commercial and use value seemed more apparent to participants than insights for research. The smart bin was viewed as a potential service for businesses to minimise their waste or akin to existing technologies, such as domestic smart meters, which monitor energy consumption.

## 7. Discussion and Concluding Remarks

Considering the pitfalls of preceding food waste analyses, the evidence presented here shows promise for crossing disciplines and methodological gaps. Where personal testimonials are self-reported and voluntary, the smart bin offers an objective means of measuring food waste, one that according to participants is enticing enough so they do not attempt to cheat, while also sufficiently passive that usage requires little extra effort. Where household studies have had no means of quantification and have experienced participants changing their behaviour to appear favourable in the eyes of the researcher, the smart bin provides evidence of consumption behaviour, in some instances undermining or revealing new insights into household practices unknown to residents [12,25,35,48].

This preliminary study has revealed a significant variety in food waste data, including in rate, weight, time, food items and avoidability, both within and between households; where patterns emerge, there are also disruptions and randomness. Whilst this is an insignificant sample size to form lasting conclusions about the social dilemma of food waste, when combining the smart bin data with the participant accounts of waste, notable instances arise of food waste as a social practice [31]. Routines and identities emerge, which seem to shape participant wasting patterns and food relationships; participants explain and defend their behaviour, as indicated by the smart bin, according to them.

There are also practical implications for mitigation initiatives, for example, the majority of waste weight being limited to a few products is potentially advantageous, allowing a focus on these key areas. A total of 90% of the items fall outside of this top group. This speaks to what people did not throw away and different ways people use the bin. Furthermore, for products that are packaged, such as yogurt, consumers may not invest the extra effort to remove the contents to compost it. Whilst these kind of assumptions cannot be fully accounted for, it is hoped that this might be included in a future iteration of this study, recording food products going into the home to understand in greater detail what people do and do not choose to throw in the compost. Combining these insights might lead to targeted interventions; leftovers are among the highest items wasted, so focusing on reducing them would make a greater impact to the sample than on other products.

When presented with the aggregated data, framed in such a way that it contextualises the information in its financial and environmental consequences, it reveals a conflict with the participant’s personal agency and desire to mitigate waste. Food waste here becomes an interplay between social and material ordering of practices; how personal agency and social structure conform to provoke a situated understanding of food waste according to an entanglement of contextualised values and resources. Therefore, the bin is seemingly able to situate considerations of food waste within the lives of users, building on culture and identity to instigate behavioural change rather than economic reasoning alone [8]. Future interventions might then move away from ‘blaming the consumer’ and develop appropriate food waste strategies according to consumer values.

This might be supported by the triangulation of insights from the literature, interview data and numerical records of consumption. For example, some of the key aggregates developed during the qualitative analysis were reflected upon, including identities, emotions and practices. The exploratory analysis suggested that identities of the provider, producer, saviour (bin picker) and (dis)enabler indicate correlation with reduced waste, whereas children and working parents are the opposite. Emotions are generally all linked to reduced waste, excluding guilt, which by a small margin was associated to increased waste. Practices mostly suggest less waste, whereas responsibilisation was with greater waste. 

The engineering of the bin manages to seamlessly reposition the action of waste into the home alongside consumption, making visible the action of food waste and serving a purpose to users. Whilst some participants described their awareness of being ‘under surveillance’, actual interaction was relatively passive, simply mimicking the traditional compost bin. However, when these came together as a smart device, relaying that information to the user, the bin provoked a strong reaction, as is intended in the approach. Participants experienced shame and anger at the information they had previously unaccounted for, but also pride in what they considered a success (lots of green waste), often leading to a shared agreement and understanding of waste practices. 

The data yielded are also considered as significant for providing a window into the home, different wasting patterns and attached consumer values. These may only become evident when aggregated or explored over time, for example, some participants would claim they had no routine, once referred to as ‘ready-steady-cook every night’; however, it was rarely recognised that this in and of itself was a routine. While this cannot be fully realised within the scope of this article, it could be suggested that those without a conscious routine are much more flexible to the need or desire to change their consumption. Those households that demonstrated strict indicators of routine (planned meals) would report greater waste as a break in their routine, either to follow a new diet, to order a takeaway or just daily pressure, and would result in a food glut, without the means to accommodate it. This speaks to both the potential to track and understand patterns and the future design of such a product.

As computing technologies grow increasingly inexpensive and as smart devices become embedded in all aspects of daily life, it does not require a great stretch of the imagination to envisage the smart bin as a domestic product. In combination with a more qualitative engagement, as requested by users and perhaps facilitated through a smart-phone app or website, the technology may aid in easing the burden of food waste, meeting consumers with possible recipes to prevent that waste from occurring again or even connecting users with channels for repurposing that food, as in the now popular application Olio. Furthermore, if these technologies were adopted on a large scale, this growing data pool could provide the basis for big data analytics, learning certain behaviours and accurately predicting the most appropriate mitigation strategy, according to a high granularity of contextual factors. 

As is traditionally the case with such devices, the greatest share of the value can be found higher up the ladder; in this case, beyond consumers and researchers and into municipalities, governments and private organisations [83]. Municipalities may be able to adapt their system of refuse collection better according to these data, predicting and pre-empting gluts in household waste to distribute labour and resources more efficiently [84]. Governments may be able to measure the impact of policy interventions, and to strategise accordingly [85]. Supermarkets and food technologists can learn about their products, how popular they are, and if they are spoiling too quickly [86]. Each of these potential uses, supported by and working in combination with a network of other smart devices, may help to build sustainable futures akin to the conceptions of a circular economy, in which highly granular processes are optimised to minimise losses and reincorporate any potential waste back into the system of value generation [87].

The next iteration of the smart bin might include a feedback system that speaks to participants, praising the proportion of not only appropriate, unavoidable waste but also healthy foods, penalising waste but understanding the values of hygiene and food provisioning while contextualising their data in familiar metrics. Table A1 is an example drawn from the user feedback sheet. It shows each avoidable item by weight, cost in British pounds sterling according to the own brand supermarket products and kilograms of carbon dioxide equivalent released from household 3. Returning to Table 1, they are significantly below the UK average in weight and cost; however, portraying the data in such a way was still highly provocative for the participants. It shows how seemingly insignificant waste, such as salad leaves or a small amount of leftovers, can over time accumulate, in this case to over a kilo of CO_2_ equivalent for each item. While this was a limitation in the current design, future iterations may integrate with advances in artificial intelligence for identifying images of waste, streamlining data processing; the bin may allow this to be achieved seamlessly, at little effort.

A further limitation, as identified by this discrepancy with the national average, may reveal the extent to which users changed their behaviour in relation to the bin, and from that we can deduce that any future design of a smart bin cannot be relied upon to paint a full picture of food waste. How people use the bin is also problematic; different people have different ideas about what to put into a food waste bin depending on what purpose that waste has, be it personal compost or refuse collection. For example, some may not want coffee grounds in their compost due to acidity.

It must also be considered that the research was conducted during COVID-19 in the UK and, due to lockdown restrictions, consumption patterns might not be as they would in usual life. It was frequently reported that being at home had led to more cooking, larger meals, and greater experimentality in cooking, all indicators of increased waste. Now, in returning to ‘normal’ life, it is expected that this will be reduced, with the majority of participants leaving the house for work on weekdays, leading to a concentration of waste in mornings, evening and weekends.

This article has attempted to make clear the shortcomings of the experiment so that future designs may be able to learn from our insights, as is part of the technology probe approach. On the one hand, the limited sample size has prevented insights into food waste on a large scale, while on the other, it has revealed the great variety within this homogenous and small sample, emphasising how the technology probe speaks to the context of use. This does not undermine the many potential insights to be drawn from the smart bin as an instrument for research. Rather, it has hinted toward a space between the understanding of food waste practices that few previous studies have been able to approach. Drawing from this short exploration, the smart bin is beginning to scratch the surface of integrated consumer behaviour to figure out what actors are doing right, based on a broad spectrum of human attributes, including culture, context, knowledge and demographic, and to use that information to build personalised, data-driven interventions [20]. This literature review and preliminary study was intended to make a case for the technology probe and then inspire further use and exploration of such methods. It is, thus, its contribution to suggest a tool that builds on the trajectory of domestic smart devices to synthesise the previous understandings of food waste and to open-up the discussion on motivators and mitigators, in the hope of developing more effective research and technologies.

## Figures and Tables

**Figure 1 foods-11-02048-f001:**
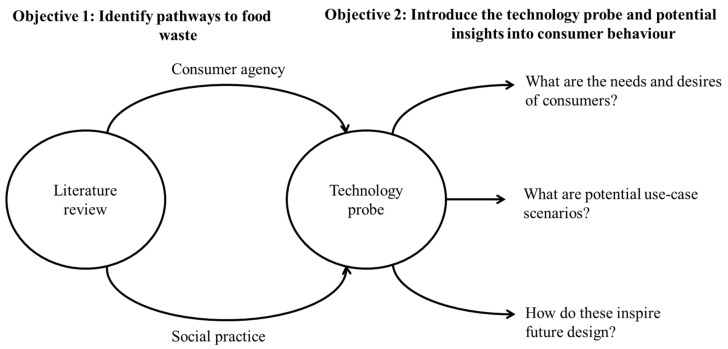
Summary of objectives.

**Figure 2 foods-11-02048-f002:**
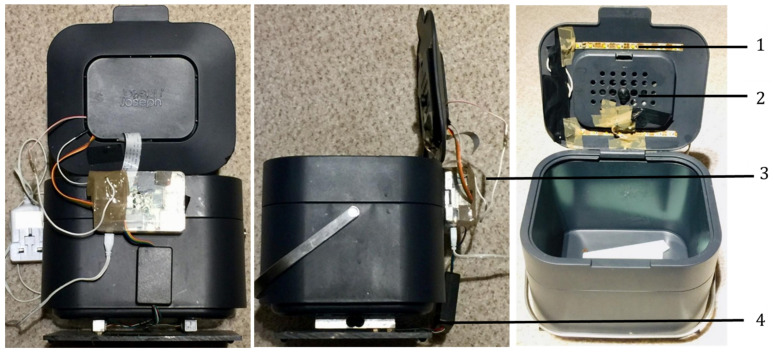
The smart bin. The smart bin is comprised of a 4-liter caddy with dimensions 23.5 × 19.7 × 16.1 cm. In the lid is a light bar (1) and a motion sensitive camera (2), that when the lid is lifted is programmed to take 6 photos. These are wired to a mains-powered, Raspberry Pi single board computer (3). This is connected to the household Wi-Fi and data is recorded remotely. Finally, the Raspberry Pi signals a set of weighing scales (4) to record the weight of the bin each time the lid is opened.

**Figure 3 foods-11-02048-f003:**
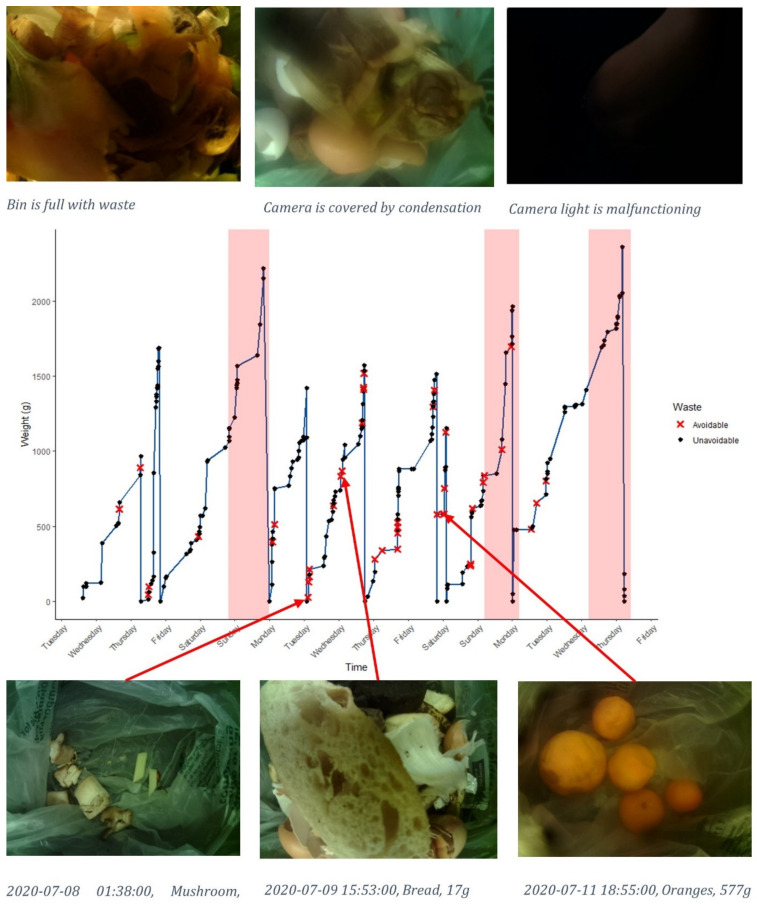
Household 3’s weight of food waste over time and photographic data.

**Figure 4 foods-11-02048-f004:**
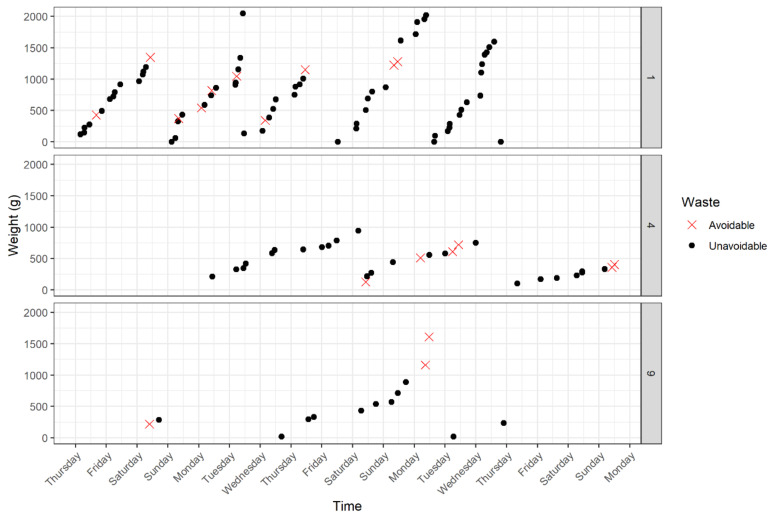
Waste over time from households 1, 4 and 9.

**Figure 5 foods-11-02048-f005:**
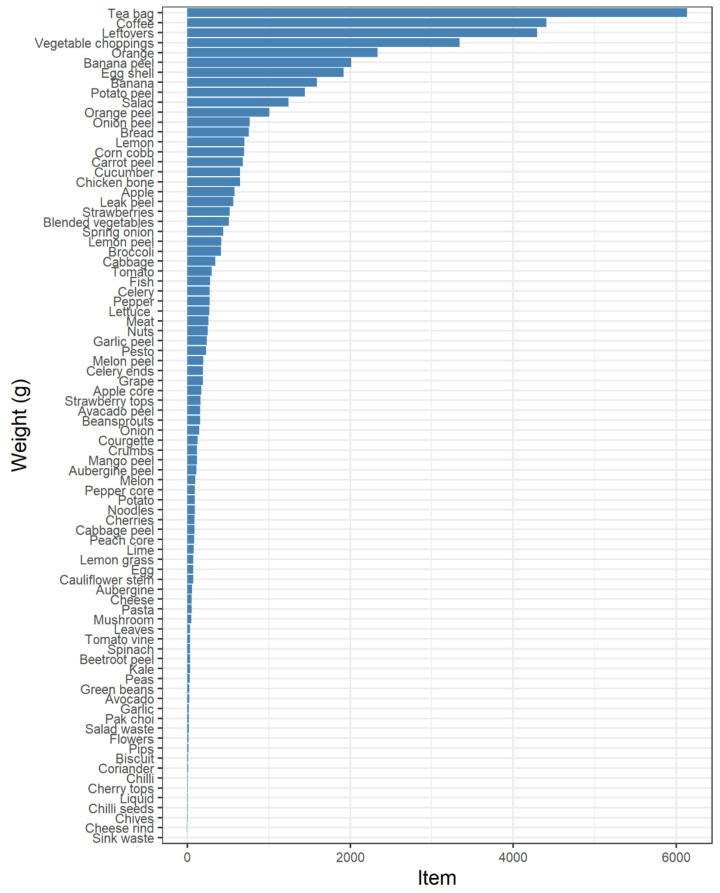
Total weight of each item.

**Figure 6 foods-11-02048-f006:**
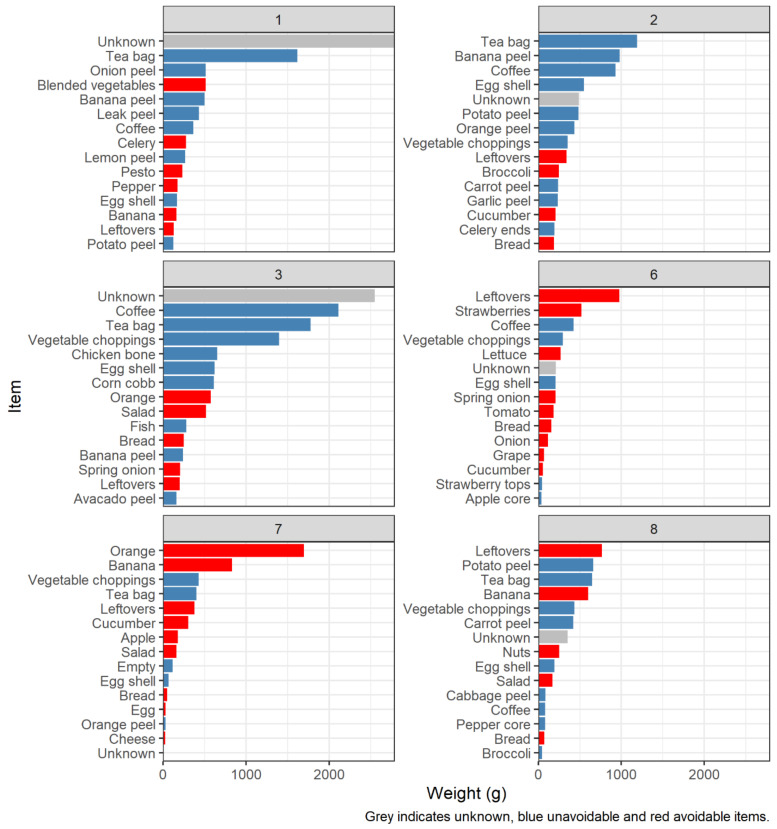
Top 10 items by households.

**Figure 7 foods-11-02048-f007:**
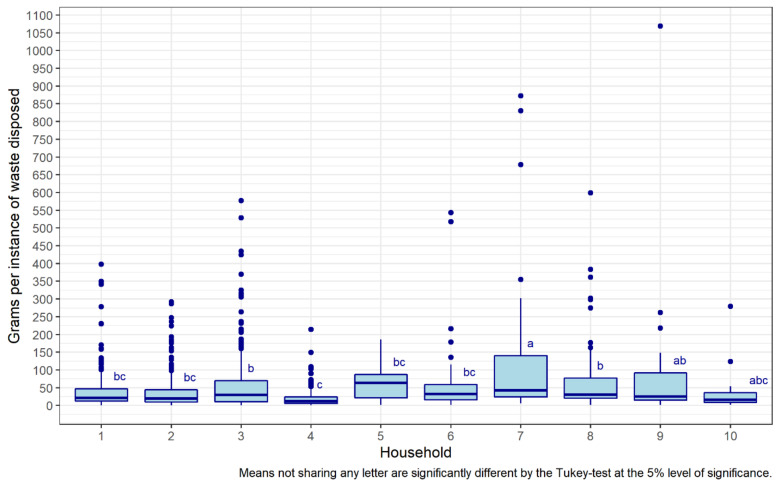
Grams per instance of waste disposed between households.

**Figure 8 foods-11-02048-f008:**
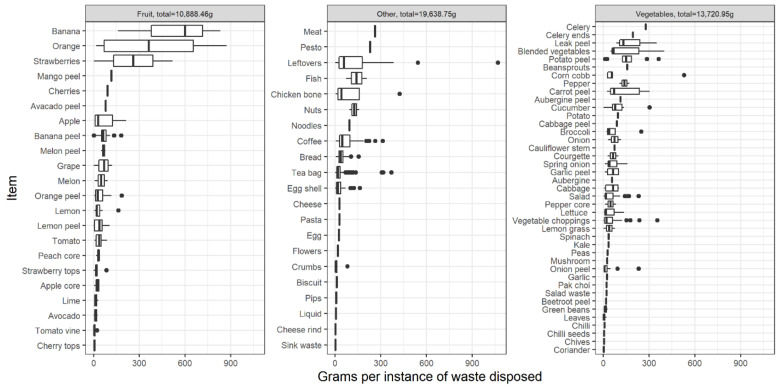
Rate of change in weight by item.

**Figure 9 foods-11-02048-f009:**
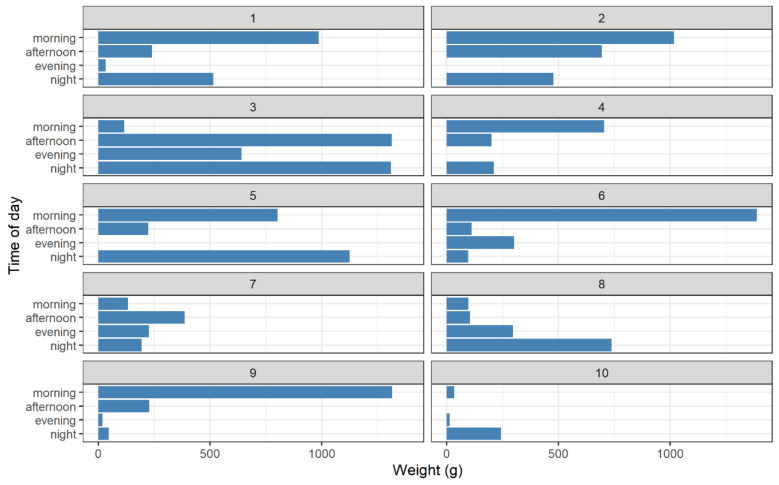
Total weight of waste and time of day across households.

**Figure 10 foods-11-02048-f010:**
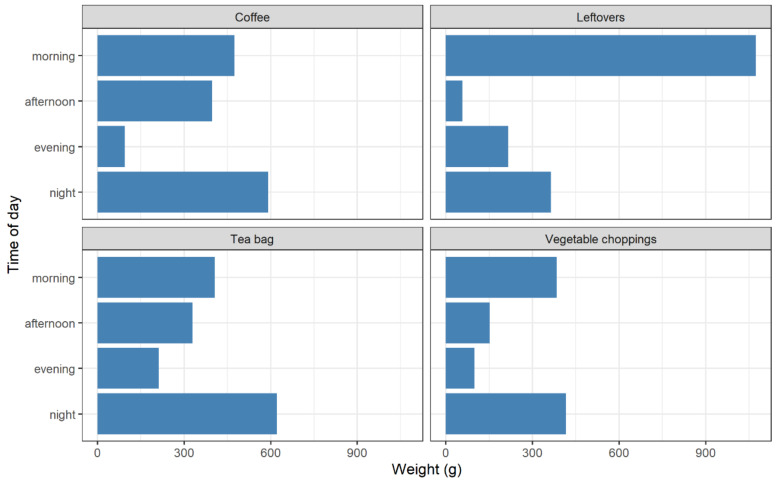
Total weight of selected waste items and time of day.

**Table 1 foods-11-02048-t001:** Descriptive statistics.

Household	Adults	Children (under 12)	Total Instances	Number of Unique Items	Avoidable Items (%)	Total Weight (g/w)	Avoidable Weight (g/w)	Avoidable Weight (%/w)	Avoidable Cost (GBP/w)	Unknown (%)
1	5	0	214	30	8	4435	907	20	5.53	35
2	4	0	226	43	13	4378	772	17	6.54	8
3	4	0	236	39	17	6758	1314	20	9.73	25
4	2	0	91	25	18	1114	257	23	2.97	3
5	1	0	35	14	23	1075	210	20	1.64	9
6	2	0	70	19	36	1896	1277	67	13.95	11
7	2	3	37	15	40	2349	1828	78	6.7	2
8	4	0	70	19	17	2470	949	39	9.75	7
9	2	0	43	14	18	1608	836	52	14.26	55
10	2	0	14	5	7	288	140	51	2.8	50
Total	28	3	1036	223	19.7	26,371	7583	29	73.87	20.5

## Data Availability

The data presented in this study are available on request from the corresponding author. The data are not publicly available due to privacy concerns and ethical commitments.

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
