# Peer review of "Consumer Behaviour and Food Waste: Understanding and Mitigating Waste with a Technology Probe"

_foods, 2022, doi:10.3390/foods11142048_

Round 1

Reviewer 1 Report

First of all we would like to stress that we have appreciated the efforts involved in this research. We acknowledge that an important amount of work has been involved in it and we certainly encourage the authors to invest themselves further in the highly relevant (and timely) topic of food waste reduction.

Yet, we have identified several flaws to the research that we would like to stress here.

1.       Objectives

The authors seem to pursue too many objectives at the same time, which renders difficult to perceive exactly what they are and what the contributions may be. For instance, it seems at first that the authors engage in a state of the art/ extensive literature review. But it quickly appears -for those who know this literature- that it is hardly exhaustive and merely corresponds to a tentative to demonstrate that the current methodologies have limits and that one should think more cross-methods to tackle the problem. I believe authors should limit their introduction to that. At this stage, it seems that authors want to run too many rabbits at the same time and the consequences are that the work seems superficial and contributions appear limited. Of course, no one wants that.

2.       Contributions

At this stage, and notwithstanding the amount of work involved, contributions seem limited. To me, the main (sole) relevant contribution relies in the demonstration of the effectiveness of the multi-method approach : adding technology to qualitative methods enables consumers to increase their awareness.

Yet, the specific approach considered here (a demanding approach combining high involvement of participants and researchers) hardly seems applicable on a large basis to increase awareness of most consumers. Who can realistic assume that consumers will equip themselves as such and that researchers will be available to discuss with them about their practices? Although it seems nice on paper, it is a not realistic at all.

Furthermore, contributions related to the quantitative analyses appear not generalizable for many reasons : sample size, sample biases, type of food waste considered (see discussion on methods for details). They also propose conclusions that are hardly usable to fight food waste (how does the time or frequency vs. quantity issue of waste contribute to solving the issue is not clear to me).

3.       Methods

a.       Selection of participants -> not much details are provided to the selection of the sample. Yet there might be an important bias in the sample, due to involvement in the issue : people may volunteer because they are already concerned by the issue-> we cannot use those results to conclude on the issue from a global population perspective. Authors do not discuss this issue.

b.       The same comment applies regarding the profile of participants. Only one household has (young) children, what impact does that have on the conclusions -> no discussion on that.

c.       Size of the sample is too limited to cover such an issue, even for a qualitative study. The problem is even more important when considering the quantitative method used to analyse some results. Furthermore, there is no control. One cannot conclude on the (reduction of ) the amount of waste  -> so we cannot really infer the impact of the bin

d.       Food waste consideration seems limited in the type of food  (mostly veggies and fruit). Those household most probably consumer meat, cheese, biscuit, bread, …. What about those as food waste? I understand – as the authors stress- that yogurt leftovers may not be added to the bin, but what about bread, biscuits? This seems to significantly reduce the amount of food waste … and the amount of relevant contributions.

e.       Analysis of waste seem hardly relevant. What do we really learn from that? It is more an info on food consumption patterns, not so much on food waste.

f.        Qualitative analyses appear too superficial. We need more insights, more verbatims, more details, etc. 

2.       Discussion

a.       Authors state that the methods previously used rely on self-reported data and this may be problematic (idea to which me agree). Yet, their own method relies to some extent to participants’ transparent/reliable behaviors. Authors do not discuss that, even of the light of some comments participants share  (they somewhat acknowledge not being 100% reliable (page 22 line 799). Authors should discuss this limit.

b.       Authors mention in the discussion issues/concepts that have not been mentioned earlier in the paper (i.e.  “identities issues”, p 27, line 1035). This is of course a problem, as we do not know neither how authors define those, nor how they consider those, what is their impact, etc.

c.       Authors should discuss more thoroughly what impact covid19 may have had on their conclusions and to what extent it may impact the current food waste issue, now that we are back to “normal” life.

d.       Authors state that their study responds to the question “to what extent does consumer agency or social practice influence individuals/household ins the adoption of sustainable eating habits” (p 27, l 1030). After a thorough reading, I’m still not sure. Authors should be more specific on the arguments they offer to answer this and in which direction this leads us

3.       Form

We invite authors to take a bit more time before submitting as those “form mistakes” may give a impression of lack of rigor.

a.       Many typos (page 9 l 447 : pe -> by, etc., p 22 l 781 : participants, ….)

b.       Structure is not systematically followed (i;e : no "pt 3.4" page 7)

Author Response

Please see the attatchment

Reviewer 2 Report

I had the opportunity to review the paper entitled "Consumer Behaviour and Food Waste: Understanding and Mitigating Waste with a Technology Probe" and I suggest a major revision in order to improve some parts of this manuscript.

There are my suggestions: 

- avoid using "we" and "our" throughout the whole text, a passive form is necessary for this type of paper

- try to change keywords that you do not have in the title, select others

- References are not cited according to the Instruction for Authors, so please revise all of it in the text and the list

- line 69 and line 101 - you do not gave two papers, so merge this to paragraphs; it is not appropriate to have two paragraphs with some aims and goals;

- In the connection with the previous comment, I need to require a graphical interpretation of goals. I think it is necessary for a better understanding of the investigation;

- In part Defining Food Waste indicate in better manner biodegradability of the food waste and its possible utilization of it.

Reviewer 3 Report

This is an interesting study and a complete work with extensive literature review and comprehensive analyses. The paper is a timely research; both clear and readable, including its structure and message. Both the methodology and approach presented would be interesting for readers working in the field, and the discussions and conclusions derived from this study are also offering some very useful insights. I do not have any further comments or suggestions for improvement of this paper. Overall, this study is an excellent contribution to the existing knowledge.

Author Response

Please see the attatchment

Round 2

Reviewer 2 Report

/